# The influence of care types on the development of orphans——An empirical study from China

Ping Liang[1], Haimei Li[1], Peng Feng[2]*

1 Humanities and Law School, Chengdu University of Technology, Chengdu, China, 2 School of Public Administration, Sichuan University, Chengdu, China

* peng_feng@scu.edu.cn

## Abstract

Care typess is considered the cornerstone of orphan's happiness and health. In 2022, China had more than 190,000 orphans in different types of care. The purpose of the study is to examine the relationship between care types and development of orphans in China. We conducted an empirical study using cross-sectional survey data related to orphans from LZ City, Sichuan Province, China, in 2020. The data sample consists of 320 valid samples of orphans and their families, including 166 boys and 154 girls. The study conducted multiple linear regression model to analyze the relationship between care types and other family variables and the development of orphans. The results indicate that compared with other care types, grandparent care has a significant positive impact on the overall development and psychological status of orphans, with this impact being moderated by the social participation of guardians. The higher the social participation of guardians, the greater the positive impact of grandparent care on the development of orphans. Further analysis, grouped by sex and age, revealed that the effects of care types on orphan development are influenced by the gender and age of orphans. Specifically, girls and younger orphans tend to exhibit better development in a grandparent care family. The research results of this study provide an empirical basis for the government to formulate policies, and for the government, society and family to jointly ensure the better development of orphans.

**Data Availability Statement:** All relevant data are within the manuscript and its Supporting Information files.

**Funding:** We are honored to have received support for our research from two funding sources:

## 1. Introduction

Orphans are a distinct group of children. They face a challenging environment with absent parental guardianship, uneven family caregiving abilities, and inadequate protection of their physical, mental, and educational rights. The effective promotion of orphans' overall development is a common concern in both society and academia. According to data from the Ministry of Civil Affairs of China, by 2020, there were 193,281 orphans in the country, including 58,989 in child welfare institutions. The number of orphans who are not raised in child welfare institutions is the largest, 134,292, accounting for 69.5% [1]. This part of orphans refers to those whose parents are deceased or whose parents have been declared dead or missing by the

ZDJS202315 and SCJJ23ND215. ZDJS202315 was a grant from "Chengdu University of Technology "Double First-Class" initiative Construction Philosophy and Social Sciences Key Construction Project(2023)", and SCJJ23ND215 was a grant from "Sichuan Provincial Philosophy and Social Science Planning Project(2023)". Professor Haimei Li was funded by the ZDJS202315 project, while Professor Feng Peng received support from the SCJJ23ND215 project. We sincerely appreciate the financial assistance provided by these funds. The funders of the study had no role in study design, data collection, data analysis, data interpretation, or writing of the report.

**Competing interests:** The authors have declared that no competing interests exist.

people's court, or who are in fact unraised and are not raised in welfare institutions and scattered in the community. According to relevant Chinese policies and regulations, the primary care types for these orphans include kinship care, institutional care, family foster care, and legal adoption [2]. In this context, it is of great reference value to study the influence of different care types on the development of orphans in order to ensure the overall development of orphans, safeguard their legitimate rights and interests, promote the formulation and improvement of relevant policies [3–6].

Currently, research on the factors influencing the overall development of orphans can be divided into three main areas. One area examines the influence of individual factors on orphans' overall development, such as their health status [7]. Research at the individual level indicates that compared to other orphans, AIDS orphans are more vulnerable physically and mentally, face greater academic challenges, and are more likely to struggle with completing their studies [8, 9].

On the one hand is the impact of policy intervention on the overall development of orphans, and the necessity of policy intervention. At present, Russia does not have a unified law on foster care. The legislation at the federal level is not fixed, and each region has its own legal regulations [10–12]. This is not conducive to the stable placement of orphans and cannot guarantee the care of foster families. Therefore, a unified federal law should be formulated to uniformly regulate the problems in the process of foster care of orphans and the standards of foster families [4]. In terms of the necessity of policy intervention, some scholars focus on the pressures and challenges faced by caregivers. According to the survey, poverty is the main obstacle for caregivers to raise orphans, and caregivers, especially grandparents, believe that the pressure on financial status is the most important to solve [13, 14]. Currently, caregivers have high economic pressure, physical pressure and emotional pressure, and poor families lack external support. Therefore, incentives to promote sustainable orphan care should focus on financial assistance, while focusing on the role of community mechanisms, assessing family care ability, and strengthening family care ability through external support and other forms of assistance [15–17]. In addition, the challenges faced by caregivers include limited resources, emotional challenges and insufficient support from other relatives. It's difficult for them to participate in organized community activities and obtain psychosocial services from community institutions. They also lack childcare skills. Therefore, caregivers should be trained to improve their care ability. At the same time, the government should promote a community-based approach, protect vulnerable groups through policies and legislation, and deliver resources to these families and communities [18]. Some scholars focus on the negative impact of institutional care on orphans and point out the necessity of policy intervention. Orphans who grow up in institutions face the risk of internalizing and externalizing problems, attention deficit hyperactivity disorder and autism. This risk is affected by the psychosocial interaction between caregivers and children. Interventions using a system-wide approach to positive behavioral support are positive for orphanage children with severe externalizing problems [19]. The negative impacts of large orphanages on children include emotional barriers, general development barriers and behavioral problems, etc. The government needs to adopt community-based and stable institutional care alternatives. At the same time, promote the transformation of large dormitory orphanages to more apartment orphanages similar to families [20]. At the same time, the government and practitioners should formulate personalized care plans for orphans, supervise and regularly review their lives after entering institutions, and formulate "suitability" standards to regulate institutional care [21].

On the other hand is the influence of different rearing types on the overall development of orphans. Different care types have different effects on the permanence and stability of orphan placement, children's mental health and behavior development. Part of the study is from the

perspective of kinship care and non-kinship foster care. The results of related studies show that children receiving kinship care are superior to traditional foster children in behavioral development, mental health, permanence and stability of placement. The traditional foster children are non relative foster children who are placed in unrelated foster parents' homes [22]. Compared with non-kinship foster care, adolescents with kinship have fewer mental health problems and perform as much as their peers in foster families, but non-kinship foster care is more likely to receive financial assistance. The government should adjust its policies to provide equivalent services to kinship care [23]. In conclusion, existing studies have shown that there is a significant difference between kinship care and non-kinship foster care [24].

In addition, some scholars conducted research on the impact of grandparent care on children's academic performance, psychological and physical health. The research points out that there are differences between the goals and strategies of grandparent care and parental care [25]. Scholars differ in their views on whether grandparent care affects children's development positively or negatively. Some scholars pointed out that grandparents have a negative impact on children's physical health and are more likely to cause childhood obesity [26]. At the same time, children raised by grandparents have no significant difference in the level of speech ability and executive function development compared with other children, but the development level of theory of mind is significantly lower than other children [27]. Some scholars believe that intergenerational education has a double-edged sword effect on children's physical and mental health rather than a single effect [28]. The quality of grandparent care will be affected by various factors such as space distance, economic situation, grandchildren, parents and grandparents. The government should formulate support policies, provide corresponding social group services, and use school and other interventions to improve the positive impact of grandparent care [29]. In terms of children's academic performance, the influence of grandparent care on children's academic performance is also affected by the individual educational ability of grandparents [30].

Based on the above studies, different parenting types have varying effects on the development of orphans, and these effects are influenced by related factors. There is no consensus on the key factors driving children's development, particularly within foster families or care institutions where orphans are placed. Lin (2018) highlighted that guardians' social participation can moderate the impact of care types on orphan development, as social participation helps buffer caregiving pressures and reduces the negative impact on children [31]. This study analyzes the social context and finds that grandparent care is the most common caregiving type among orphans. Does grandparent care have a more positive impact on orphan development? Is this effect moderated by other factors? To address these questions, this study builds on Lin's (2018) research and, while considering the moderating effect of guardians' social participation, examines the impact of different care types, including grandparent care, on the development of orphans [32–34].

This study uses the questionnaire survey data of orphans from Luzhou Municipal Civil Affairs Bureau in 2021, adopts a linear regression model to conduct empirical research, and focuses on the impact of care types on the development of orphans under the moderating effect of social participation of guardians. This study is conducive to enriching the understanding of the factors affecting the development of orphans, helping the government to formulate and improve relevant policies, improving the positive impact of grandparent care on the development of orphans, and coordinating and supporting the government, society and family to jointly ensure the well-being of orphans. The research contents are arranged as follows: The first part is an introduction, the second part is research methods, the third part is empirical results analysis, the fourth part is discussion, and the fifth part is conclusion analysis.

## 2. Materials and methods

### 2.1 Data and sample

The data used in this study are from the questionnaire survey data of 545 children randomly sampled from the orphan system of Luzhou Municipal Civil Affairs Bureau in 2021 and the matching data of the comprehensive query data of orphans and disabled children of Luzhou Municipal Civil Affairs Bureau. After matching the information of placement type to the questionnaire data, there are 372 samples left.

The specific questionnaire survey was conducted by PPS method, which was stratified, multi-stage and proportional to the scale. The Luzhou Municipal Civil Affairs Bureau organized investigators who completed the relevant training to conduct household survey. When filling in the questionnaire, the investigators were responsible for guidance and quality control. Some children under 10 years old who need to fill in the questionnaire should be answered by their guardians, while those aged 10–18 years old should answer their own questions. The investigation has been approved by the guardians in advance, and the investigation process is in line with research ethics.

Since this study is mainly based on the moderating effect of guardians' social participation to study the impact of parenting types on the development of orphans, after excluding the missing values of care types, individual characteristics data of orphans and their guardians, and family care capacity data, 320 samples were retained.

### 2.2 Measures

**Development of minors.**   The development of minors must be multi-dimensional, so the indicators reflecting the development of orphans must be multi-dimensional. We set up indicators from two different dimensions, including learning and psychological situations of minors. If the minor's learning situation is good, it will be coded as 1, and if it is bad, it will be coded as 0; If the minor's psychological situation is good, it is coded as 1, and if it is bad, it is coded as 0. At the same time, this study combined these two dichotomous variables into a general indicator to reflect the overall development of minors. The value range is 0–2. The larger the value, the better the overall development of orphans.

**Type of placement.**   In the comprehensive query data of Luzhou Municipal Civil Affairs Bureau for orphans and disabled children, the types of placement for orphans mainly include grandparent care, elder brother and elder sister care, other relatives care, non-kinship care and so on. This study mainly focuses on the impact of grandparent care and other care types on orphans. Therefore, in the empirical analysis, this study generates a dummy variable. If the placement type of orphans is grandparent care, it is coded as 1. If the placement type of orphans is the placement type other than grandparent care, it is coded as 0, as the benchmark category [35].

**Guardians' social participation.**   The social participation of guardians refers to the ability and degree of individual participation in social activities and social relations. Good social participation means that guardians have good physical and mental health, upbringing patterns and rearing behavior. This study refers to the approach of Lin (2018) and studies the influence of care types on the development of orphans based on the moderating effect of guardians' social participation. In the aspect of measuring social participation, this study synthesizes the indicators of social participation ability from seven dimensions: whether guardians participate in public welfare activities, whether they abide by social morality and have good personal conduct, whether they abide by national laws and regulations and related systems, which can reflect the social participation of guardians. In the process of coding, yes is encoded as 1,

**Table 1. Variable description table.**

| | Variable name | Variable description | Value range |
|---|---|---|---|
| Dependent variable | Overall development | The larger the value is, the better the overall development is. | 0–2 |
| | Learning situation | 1 is good, 0 is bad. | 0–1 |
| | Psychological situation | 1 is good, 0 is bad. | 0–1 |
| Independent variable | Placement type | 1 for grandparent care, 0 for other types of placement. | / |
| Characteristics of minors | Gender | 1 for boys and 0 for girls | / |
| | Age | / | 0–18 |
| | Educational level of minors | 0 is illiterate, 1 is kindergarten and preschool, 2 is dropout, 3 is primary school, 4 is junior high school, 5 is high school, 6 is university | 0–6 |
| Characteristics of guardians | Guardian gender | 1 is male and 0 is female | / |
| | Guardian age | / | > = 18 |
| | Guardian marriage | 1 is married and 0 is single | / |
| | Educational level of guardians | 0 is illiterate, 1 is kindergarten and preschool, 2 is dropout, 3 is primary school, 4 is junior high school, 5 is high school, 6 is university. | 0–6 |
| | Health status of guardians | 1 is healthy, 0 is unhealthy. | / |
| Family economic situation | Family economic situation | 1 is good, 0 is bad. | / |
| Family care ability | Living conditions | Composed of seven indicators, the smaller the value, the better the living conditions. | 7–21 |
| | Family hygiene | Composed of one indicator, the smaller the value is, the better the family hygiene is. | 1–3 |
| | Family relationship | Composed of four indicators, the smaller the value, the better the family relationship. | 4–12 |
| | Social participation ability | Composed of seven indicators, the smaller the value, the better the ability to participate in society. | 7–21 |
| | Living care ability | Composed of one indicator, the smaller the value, the better the ability to care for life. | 1–3 |
| | Family social relations ability | Composed of 12 indicators, the smaller the value, the better the ability of family social relations. | 12–36 |
| | Family resource acquisition ability | It is composed of one indicator, and the smaller the value, the better the family resource acquisition ability. | 1–3 |

uncertainty is encoded as 2 and no is encoded as 3, so the index of social participation ability is 7–21, and the smaller the value, the better the social participation of the guardians.

**Control variables.** In the choice of control variables, we choose from four aspects: characteristics of minors, characteristics of guardians, family economic situation and family care ability. This study refers to the practices of other scholars and takes into account the characteristics of minors and guardians [36]. Among them, the characteristics of minors control their gender, age and educational level. In terms of guardian characteristics, the guardian's gender, age, marriage, educational level, and whether the guardian has major diseases or chronic diseases shall be controlled. At the same time, the family characteristics of guardians will also affect the development of minors [37]. Therefore, we incorporate it into the control variables. We measure the family characteristics of guardians by family economic situation and family care ability. In terms of family economic situation, we choose whether the family has a stable income as the control variable. In the aspect of family care ability, we choose family living conditions, family hygiene, family relationship, social participation, living care ability, family social relations ability, family resources acquisition ability as control variables. The variable value description is shown in Table 1.

## 2.3 Empirical model

In this study, based on the moderating effect of guardians' social participation, this study uses the empirical model to test the influence of care types on the development of orphans, that is,

compared with other types of placement, whether grandparent care will have a positive impact on the development of orphans. The basic model is as follows:

$$CSW_i = \alpha + \beta PT_i + \gamma PT_i * SE_i + \delta X_i + \varepsilon_i$$

Among them, $CSW_i$ is the dependent variable representing child well-being, $\alpha$ is a constant. $PT_i$ is the explanatory variable of interest in this study, representing the placement type of child $i$. Based on existing research, this study introduces the interaction term $PT_i * SE_i$ to research the moderating effect of $SE_i$, $SE_i$ represents the social participation of guardians corresponding to child $i$. $X_i$, represents a series of control variables that may affect the development of children, including minor characteristics, guardian characteristics, family economic situation and family care ability. $\varepsilon_i$ is the error term.

## 2.4 Analytic method

In this study, STATA software is used for statistical analysis, and the OLS multiple regression method is used to explain the research problems. Because the development of orphans in this study is a multi-dimensional index, in order to explore the impact of care types on all aspects of the development of orphans, this study includes different dependent variables into multiple regression models in turn. In addition, because this study is based on the moderating effect of guardians' social participation, in the analysis, we add the interaction term of care types and social participation variables as moderating factors to the multiple regression model for OLS regression.

## 3. Results

### 3.1 Descriptive results

The descriptive statistical results of the relevant variables in this study are as follows(see Table 2). Among them, the average independent variable placement type is 0.525, indicating that most of the 320 samples of orphans in Luzhou City are raised by grandparents. Among the dependent variables, the average value of the overall development of minors is 1.423, that is, on the whole, the development of orphans in Luzhou City is good, in the middle and upper level. From the perspective of various dimensions of orphans' development, the average scores of learning and psychological situation of orphans in Luzhou City are more than 0.5, indicating that the development of orphans in Luzhou City is good in these two dimensions, and it is in the middle and upper level. In terms of control variables, from the characteristics of minors, the sample includes orphans from kindergarten to high school, in which the number of boys is slightly higher than that of girls; from the characteristics of guardians, we can see that the average age of guardians is 53.3 years old, combined with the standard age classification standard of the World Health Organization, we can see that guardians are generally middle-aged and elderly. In terms of family economic situation, the average sample is 0.694, that is, among the guardian families of orphans in Luzhou City, most of them have a stable income, but some families still do not have a stable income. In terms of family care ability, according to the descriptive statistical results, on the whole, the guardian families of orphans in Luzhou City are at a good level in terms of living conditions, family hygiene, family relationship, social participation ability, living care ability, family social relations ability, family resources acquisition ability and so on.

### 3.2 The moderating effect of guardians' social participation

Based on the moderating effect of guardians' social participation, Table 3 reports the impact of the type of placement on the development of orphans. Table 3 (1) reports the impact of care

**Table 2. Descriptive statistics of variables.**

|  | Variable | (1) Number of Samples | (2) Average | (3) Standard Deviation | (4) Minimum value | (5) Maximum value |
|---|---|---|---|---|---|---|
| Dependent variable | Overall development | 318 | 1.423 | 0.759 | 0 | 2 |
|  | Learning situation | 320 | 0.644 | 0.480 | 0 | 1 |
|  | Psychological situation | 318 | 0.786 | 0.411 | 0 | 1 |
| Independent variable | Placement type | 320 | 0.525 | 0.501 | 0 | 1 |
| Characteristics of minors | Gender | 320 | 0.519 | 0.501 | 0 | 1 |
|  | Age | 320 | 14.65 | 2.501 | 7 | 18 |
|  | Educational level of minors | 320 | 4.106 | 0.813 | 1 | 5 |
| Characteristics of guardians | Guardian gender | 314 | 0.573 | 0.496 | 0 | 1 |
|  | Guardian age | 308 | 53.30 | 17.17 | 20 | 85 |
|  | Guardian marriage | 314 | 0.745 | 0.437 | 0 | 1 |
|  | Educational level of guardians | 282 | 3.404 | 1.121 | 0 | 6 |
|  | Health status of guardians | 320 | 0.963 | 0.191 | 0 | 1 |
| Family economic situation | Family economic situation | 320 | 0.694 | 0.462 | 0 | 1 |
| Family care ability | Living conditions | 308 | 7.539 | 1.314 | 7 | 14 |
|  | Family hygiene | 316 | 1.051 | 0.272 | 1 | 3 |
|  | Family relationship | 226 | 4.602 | 1.340 | 4 | 11 |
|  | Social participation ability | 310 | 7.645 | 1.257 | 7 | 14 |
|  | Living care ability | 314 | 1.070 | 0.280 | 1 | 3 |
|  | Family social relations ability | 308 | 13.68 | 1.632 | 12 | 22 |
|  | Family resource acquisition ability | 316 | 1.044 | 0.261 | 1 | 3 |

types on the overall development of orphans, and (2) and (3) respectively reports the impact of care types on the learning and psychological situation of orphans.

Specifically, the influence of the interaction term of placement type and social participation ability on the overall development and psychological situation of orphans was significantly positive in the statistical sense of 1%. It indicates that after controlling the control variables at the level of child characteristics, guardian characteristics, family economic situation and family care ability, grandparent care has a significant positive impact on the overall development and psychological situation of orphans, and this influence is regulated by the social participation of guardians. In this study, social participation is a negative index, the greater the value of social participation, the worse the social participation of guardians. Combined with the regression results, it can be found that with the decrease of grandparents' social participation, the positive impact of grandparent care on children's development is weakened, and with the increase of grandparents' social participation, the positive impact of grandparent care on children's development is strengthened. The influence of the interaction term of placement type and social participation ability on the learning situation of orphans is not significant, which is due to the fact that the educational level of guardians is generally low in the sample. According to descriptive statistics, the average educational level of guardians is 3.404, which was between primary school and junior middle school. And the samples are from the same province and city of the rural areas, there is little difference in the quality of education.

## 3.3 Heterogeneous impact analysis

To enhance the robustness of our findings, we performed a heterogeneity analysis. Following the methodology outlined in previous studies [38], we conducted a subgroup analysis of orphans based on gender and age. This section analyzes the heterogeneity from two aspects of

**Table 3. Influence of care types on the development of orphans.**

| Variable | (1) Overall development | (2) Learning situation | (6) Psychological situation |
|---|---|---|---|
| Placement type | 2.485*** | 0.787 | 1.528*** |
| | (3.48) | (1.57) | (4.25) |
| Placement type * Social participation ability | -0.322*** | -0.100 | -0.195*** |
| | (-3.52) | (-1.56) | (-4.24) |
| Gender | -0.033 | -0.060 | 0.024 |
| | (-0.23) | (-0.62) | (0.30) |
| Age | -0.035 | -0.038* | -0.002 |
| | (-1.12) | (-1.73) | (-0.14) |
| Educational level of minors | 0.136 | 0.142* | 0.015 |
| | (1.36) | (1.90) | (0.30) |
| Guardian gender | 0.047 | -0.041 | 0.072 |
| | (0.30) | (-0.42) | (0.88) |
| Guardian age | -0.002 | -0.000 | -0.003 |
| | (-0.39) | (-0.06) | (-0.92) |
| Guardian marriage | 0.197 | 0.159 | 0.091 |
| | (1.12) | (1.39) | (0.92) |
| Educational level of guardians | 0.001 | 0.011 | -0.015 |
| | (0.02) | (0.19) | (-0.35) |
| Health status of guardians | -0.083 | 0.054 | -0.150 |
| | (-0.45) | (0.27) | (-1.19) |
| Family economic situation | 0.244 | 0.155 | 0.045 |
| | (1.50) | (1.52) | (0.51) |
| Living conditions | -0.058 | 0.009 | -0.043 |
| | (-0.82) | (0.19) | (-1.25) |
| Family hygiene | -0.397 | -0.125 | 0.232* |
| | (-1.65) | (-0.44) | (1.94) |
| Family relationship | -0.130* | -0.065 | -0.067 |
| | (-1.84) | (-1.31) | (-1.22) |
| Social participation ability | 0.046 | -0.031 | 0.056 |
| | (0.63) | (-0.55) | (1.55) |
| Living care ability | -0.469** | -0.185 | -0.266** |
| | (-2.05) | (-1.41) | (-2.08) |
| Family social relations ability | -0.056 | -0.008 | -0.020 |
| | (-0.98) | (-0.22) | (-0.70) |
| Family resource acquisition ability | -0.977*** | -0.445*** | -0.508*** |
| | (-6.61) | (-4.22) | (-7.16) |
| Constant | 18.946** | 8.924 | 9.324 |
| | (2.43) | (1.61) | (1.53) |
| Observations | 202 | 204 | 202 |
| R-squared | 0.416 | 0.313 | 0.380 |

Note

***, ** and * indicate that they are significant at the level of 1%, 5% and 10% respectively; the control variables include the characteristics of minors, guardians, family economic situation and family care ability.

Table 4. Impact of care types on the development of orphans of different genders and levels of education.

| Variable | (1) Girl | (2) Boy. | (3) Primary and junior high school stage | (4) Senior high school stage |
|---|---|---|---|---|
| Care type | 4.885*** | 1.004 | 2.485*** | 2.186** |
| | (3.01) | (0.85) | (3.48) | (2.29) |
| Care type * Social participation ability | -0.672*** | -0.123 | -0.322*** | -0.287** |
| | (-3.11) | (-0.95) | (-3.52) | (-2.30) |
| Characteristics of minors | Control | Control | Control | Control |
| Characteristics of guardians | Control | Control | Control | Control |
| Family economic situation | Control | Control | Control | Control |
| Family care ability | Control | Control | Control | Control |
| Constant | 17.861 | 5.172 | 18.946** | 1.883 |
| | (1.25) | (0.41) | (2.43) | (0.07) |
| Observations | 100 | 102 | 202 | 72 |
| R-squared | 0.561 | 0.472 | 0.416 | 0.508 |

Note

***, **and * are significant at the levels of 1%, 5% and 10%, respectively.

gender and educational level, taking into account the moderating effect of guardians' social participation, while testing whether there are gender differences and educational level differences in the influence of care types on the development of orphans. In the specific operation process, considering the length of this study and the focus of the study, this study only returns to the overall development of orphans.

The regression results are shown in Table 4, models(1)-(2). According to the results of heterogeneity analysis of gender dimensions, it can be concluded that grandparent care has a positive impact on the development of boys and girls, but the impact on boys is small and not significant, and the impact on girls is significant and statistically significant. To further assess the robustness of our conclusion, we introduced social participation as a moderating variable for caregiving. The regression results remain significant when social participation is included as an interaction term.

Models (3)-(4) reports the results of heterogeneity analysis of educational level. In China's education system, the nine-year compulsory education system is an important part. It is a compulsory education system for school-age children and adolescents for a certain number of years in accordance with the provisions of the law. It includes two stages: primary school and junior high school. It is free, universal and compulsory. This study takes the compulsory education stage as the division standard, divides the sample into "primary school and junior high school stage" and "senior high school stage" two groups. After regression to the overall situation of the development of orphans, the regression results shows that the types of placement have a positive impact on the overall development of minors in compulsory education and non-compulsory education, and the significant levels were 1% and 5%, respectively. Among them, the impact on minors in primary and junior high school is higher than that on minors in senior high school.

## 4. Discussion

This paper supports the exploration of the relationship between care types and orphan development. We examined the impact of different care types on orphan development, as well as

other factors influencing their growth across various care settings and the differences among them. The research findings indicate the following.

First, the results of this study show that the social participation of guardians has a moderating effect on the influence of care types on the development of orphans. The reason for this result may be that, compared with other types of care, grandparent care ensures the continuity of family upbringing and the need for children's emotional development, can alleviate children's behavioral and psychological problems, and reduce their anxiety level. At the same time, grandparents have more time and energy than other types of guardians, can better accompany minors, and grandparents have more experience in life, can guide children, improve their social adaptability [39–42]. Social participation refers to the ability and degree of individual participation in social activities and social relations. Good social participation ability means good physical and mental health, upbringing patterns and upbringing behavior of guardians. Social participation plays an important role in the social isolation and vulnerable group of orphans, and can promote the change of family resilience [43]. Therefore, when grandparents' social participation is increased, its positive impact on the development of orphans will also be strengthened.

Second, compared with boys, care types has a greater impact on girls' development. The reason, combined with existing studies, may be explained as follows: on the one hand, girls care more about emotional quality than boys, their emotional needs are more sensitive than boys [44]. Grandparent care ensures the continuity of family upbringing and the need for emotional development of minors [45]. So compared with boys, grandparent care has a significant positive impact on the overall development of girls, and the impact is significant. On the other hand, it can be explained in combination with the rule internalization of minors proposed by Wang Su [46]. The internalization of rules refers to the process in which individuals accept social norms, especially guardians' values, and turn them into their own codes of conduct. In areas related to social participation, such as morality and custom, girls have a higher level of rule internalization than boys, so when guardians have high social participation, girls also have a higher level of rule internalization, which will have the moderating effect and make girls have a greater positive impact on the grandparent care [18, 47, 48]. In addition, the level of rule internalization of girls is more likely than that of boys to be affected by the quality of upbringing. Positive parental behavior and parent-child relationships promote the internalization of girls' rules. When the guardian's social participation is high, it means that guardians have good and positive upbringing styles and behaviors, which will strengthen the internalization of girls' rules and, in turn, strengthen the positive impact of grandparent care. Therefore, in the regression analysis of girls, the intersection multiplier is very significant, revealing the moderating effect of guardians' social participation [49].

Third, for orphans at different stages of education, the influence of care tpyes has obvious heterogeneity. The possible reason is that the minors in primary and junior high school are in the stage of compulsory education, their minds are not yet mature, they still have obvious characteristics of children, and in the stage of compulsory education, minors get along with guardians for more time and are more closely related, minors are dependent on guardians, have a single understanding of things, and are more vulnerable to the influence of guardians and their social participation. The senior high school stage belongs to the non-compulsory education stage, at this stage, the time of minors in school increases suddenly, the time to get along with the guardian decreases, the dependence on the guardian weakens, the mind gradually matures, the dialectical thinking ability has been improved, the personality tends to be independent and the social network is also gradually established, the future trend also begins to appear the big difference. At this stage, minor more depends on the companion and own social

network. Therefore, children in compulsory education are more vulnerable to grandparents and their social participation than children in non-compulsory education [50].

## 4.1 Strengths and limitations

In this study, we mainly provide two contributions. First of all, based on the moderating effect of guardians' social participation, this paper studies the influence of care types on the development of orphans, which is an important supplement to the literature on the influence of care types on the development of orphans. Secondly, this study also analyzes the heterogeneity of orphans from different educational levels and genders, and further discusses the influence of care types on the development of orphans with different characteristics under the moderating effect of guardians' social participation, which is helpful to fully understand the role of care types in the development of orphans. This is conducive to governments to better formulate policies to ensure the good development of orphans.

There are also some shortcomings. First, due to the limitations of data, the conclusions of this study only explore the impact of the types of placement based on the social participation of guardians on orphans and the gender and educational level differences of this impact, and can not comprehensively analyze more impact mechanisms. Second, this study focuses on the impact of grandparent care on the development of orphans, which is different from other types of placement. There is no detailed study on the influence of each type of placement, such as elder brother and elder sister care and non-kinship care, which is related to the small number of corresponding samples. In the next step, we can further expand the scope of research, expand the number of samples, so as to carry out richer mechanism research and improve the representativeness of the study.

## 4.2 Policy implications

According to the research results of this paper, we put forward some suggestions for reference, which have certain policy significance for ensuring the good development of orphans from the government, social and family levels and improving the child welfare security system in China. First, at the government level, we should adhere to the implementation of the relevant policy documents on the placement channels of orphans, protect the rights of grandparents as the first guardians of orphans. At the same time, taking into account the family economic situation and labour force situation of grandparents, the government should provide more support, such as subsidies, to families raised by grandparents. Second, at the social level, considering that kinship caregivers have fewer resources and receive less training, services and support [51], and the upbringing of grandparents is lack of support and services to help them achieve effective upbringing. Communities, schools and social organizations should actively provide support and assistance to grandparents, such as intervention in the generation gap between older caregivers and minors to ensure effective communication between them. At the same time, in view of the low level of education of grandparents, which makes it difficult to effectively meet the educational needs of children, communities, schools and social organizations should provide assistance such as homework in response to the learning needs of children. In addition, this study points out that the influence of care types on the development of orphans is regulated by the social participation of guardians. Therefore, society should also create conditions to encourage and improve the social participation of grandparents. Third, at the family level, grandparents should improve the ability of family care from the aspects of living conditions, family hygiene, social participation ability, family social relations ability, family resources acquisition ability and so on, so as to provide a better care environment for orphans.

## 5. Conclusions

By studying the impact of care types on the development of orphans in China, we found that grandparent care can promote the development of orphans. We also found that this impact was moderated by the social participation of guardians. The higher the social participation of guardians, the greater the positive impact of grandparent care on the development of orphans. Moreover, empirical research suggested that girls and younger orphans tend to exhibit better development in a grandparent care family.

Grandparent care is the predominant type of care for orphans' care in China. Considering the economic situation and labor capacity of grandparents, more support, such as subsidies, should be provided to families where grandparents are the primary caregivers. The government should also ensure the implementation of relevant policy documents on orphan placement and protect the rights of grandparents as primary guardians.

## Supporting information

**S1 Dataset.**
(XLSX)

## Acknowledgments

**Disclaimer/publisher's note:** The statements, opinions and data contained in all publications are solely those of the individual author(s) and contributor(s) and not of MDPI and/or the editor(s). MDPI and/or the editor(s) disclaim responsibility for any injury to people or property resulting from any ideas, methods, instructions or products referred to in the content.

## Author Contributions

**Conceptualization:** Peng Feng.

**Data curation:** Haimei Li.

**Writing – original draft:** Ping Liang.

**Writing – review & editing:** Peng Feng.

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
