## [Decision Letter · Decision Letter 0]

30 May 2024

PONE-D-24-15042The influence of placement types on the Development of Orphans——Moderating effect based on guardian's social participationPLOS ONE

Dear Dr. Peng,

Thank you for submitting your manuscript to PLOS ONE. After careful consideration, we feel that it has merit but does not fully meet PLOS ONE’s publication criteria as it currently stands. Therefore, we invite you to submit a revised version of the manuscript that addresses the points raised during the review process.

Please submit your revised manuscript by Jul 14 2024 11:59PM with all revisions as suggested by reviewers. If you will need more time than this to complete your revisions, please reply to this message or contact the journal office at plosone@plos.org. Please include the following items when submitting your revised manuscript:A rebuttal letter that responds to each point raised by the academic editor and reviewer(s). You should upload this letter as a separate file labeled 'Response to Reviewers'.A marked-up copy of your manuscript that highlights changes made to the original version. You should upload this as a separate file labeled 'Revised Manuscript with Track Changes'.An unmarked version of your revised paper without tracked changes. You should upload this as a separate file labeled 'Manuscript'.

We look forward to receiving your revised manuscript.

Kind regards,

Satabdi Mitra, M.D(Community Medicine )

Academic Editor

PLOS ONE

“Funded by Chengdu University of Technology "Double First-Class" initiative Construction Philosophy and Social Sciences Key Construction Project（ZDJS202315）

and Sichuan Provincial Philosophy and Social Science Planning Project（SCJJ23ND215）”

Reviewers' comments:

Reviewer's Responses to Questions

**Comments to the Author**

1. Is the manuscript technically sound, and do the data support the conclusions?

Reviewer #1: Yes

Reviewer #2: Partly

2. Has the statistical analysis been performed appropriately and rigorously? 

Reviewer #1: Yes

Reviewer #2: I Don't Know

3. Have the authors made all data underlying the findings in their manuscript fully available?

Reviewer #1: Yes

Reviewer #2: Yes

4. Is the manuscript presented in an intelligible fashion and written in standard English?

Reviewer #1: Yes

Reviewer #2: Yes

5. Review Comments to the Author

Reviewer #1: This is a thorough review and all the statistcial analysis data has been clearly presented. It gives a good sense of all the parameters used to perform the heterogenous analysis on different aspects of welfare and has not just focused on social participation.

Reviewer #2: Abstract

1. Improve the abstract structure that contains the essence of the problem, research objectives, methodology, population & sample, data collection techniques using measuring instruments, data analysis techniques, results, conclusions, and recommendations.

2. Grammar and Style:

a. Change "placement types is a key factor" to "placement types are a key factor".

b. Change "grandparent care have a significant positive impact" to "grandparent care has a significant positive impact".

c. The sentence "and this impact is moderated by the social participation of guardians" could be clearer if rephrased to "with this impact being moderated by the social participation of guardians."

Detailing the Methodology:

a. Mention the sample size or provide a brief description of the survey used to give clearer context about the data.

b. Specify the data analysis techniques used, such as regression analysis, to add depth to the information.

Linking Sentences:

Improve the connection between sentences. For example, "Based on the moderating effect of guardians' social participation" could be clarified how it leads to "this paper uses the survey data...".

Specifying Heterogeneity Tests:

Expand a bit on the "Heterogeneity tests" to explain why these tests were conducted and what was specifically measured.

Introduction

1. In lines 30-32, that is According to China's relevant policies and regulations, the placement types for these orphans mainly include kinship care, institutional care, family foster care and legal adoption. Can you further explain what is meant by kinship care, institutional care, family care and legal adoption.

2. State the main objectives of the research. What does the researcher aim to achieve with this study?

3. Explain why this issue is important and identify the gaps or shortcomings in previous research that this study aims to address.

Method

Research methods contain information ranging from research design, population and sample, research variabels, research tools and instruments, data collection procedures, data analysis and research ethics.

1. Research Design:

Describe the type of research conducted, Mention whether the research is qualitative, quantitative, or mixed.

2. Population and Sample:

Describe the population of the study. Explain how the sample was selected and provide information about the sample size. State the inclusion and exclusion criteria used to select the sample.

3. Research variables, explain what the research variables are and their operational definitions.

4. Research Instruments:

Describe the tools and instruments used to collect data and provide information on the validity and reliability of these instruments.

5. Data Collection Procedures:

Describe the steps taken to collect data, including when and where the data were collected. Explain if any special procedures or techniques were used.

6. Data Analysis:

State the data analysis techniques used and describe the software or tools used for data analysis.

7. Ethical Considerations:

Describe the ethical considerations taken, including ethics committee approval, written informed consent from participants, and data confidentiality.

Discussion

1. In the discussion, avoid writing the table as in Table 4 Impact of placement type on the development of orphans of different genders and education levels

2. Discuss how your results relate to findings from previous relevant research. Identify any similarities, differences or contradictions between your results and previous research.

Conclusions

The conclusion has clearly presented the main findings of the research. However, it needs to be revised to improve clarity and consistency to make it easier for readers to understand the main findings and implications of your research.

6. PLOS authors have the option to publish the peer review history of their article (what does this mean?). If published, this will include your full peer review and any attached files.

Reviewer #1: **Yes: **Eeshika Mitra

Reviewer #2: No

---

## [Author Response · Author response to Decision Letter 0]

15 Aug 2024

Response to Reviewer#1:

This is a thorough review and all the statistical analysis data has been clearly presented. It gives a good sense of all the parameters used to perform the heterogenous analysis on different aspects of welfare and has not just focused on social participation. 

We express our sincere gratitude to the reviewer for providing insightful comments on our paper. Your feedback has been invaluable in enhancing the overall quality of this study. We have thoroughly adjusted various parts of the article and all modified parts are marked in red.

Response to reviewer#2:

Abstract

1. Improve the abstract structure that contains the essence of the problem, research objectives, methodology, population & sample, data collection techniques using measuring instruments, data analysis techniques, results, conclusions, and recommendations.

2. Grammar and Style:

a. Change "placement types is a key factor" to "placement types are a key factor".

b. Change "grandparent care have a significant positive impact" to "grandparent care has a significant positive impact".

c. The sentence "and this impact is moderated by the social participation of guardians" could be clearer if rephrased to "with this impact being moderated by the social participation of guardians."

Thank you for providing valuable suggestions for refining the abstract of the article. These changes will enhance the article's alignment with current literature and improve its overall standardization.

Firstly, we've adopted your suggestion and rewrite the whole abstract according to the sequence of the essence of the problem, research objectives, methodology, population & sample, data collection techniques using measuring instruments, data analysis techniques, results, conclusions, and recommendations. All modified parts are marked in red(lines 9-21)

Secondly, we appreciate your attention to grammar and style problem. We have changed all the three expression errors and marked in red (lines 9,15,16).

Detailing the Methodology:

a. Mention the sample size or provide a brief description of the survey used to give clearer context about the data.

We thank the reviewer for this constructive suggestion. We have added a brief description of survey and marked it in red（lines 11-13）. The data sample consists of 320 valid samples of orphans and their families, including 166 boys and 154 girls. And，the data we used was based on cross-sectional survey related to orphans from LZ City, Sichuan Province, in 2020.

b. Specify the data analysis techniques used, such as regression analysis, to add depth to the information.

Thank you for your thorough review of the abstract and your valuable comments. We have rephrased the sentence and marked it in red（lines 13）to explain the analysis techniques we used. The study conducted multiple linear regression model to analyze the relationship between care types and other family variables.

Linking Sentences:

Improve the connection between sentences. For example, "Based on the moderating effect of guardians' social participation" could be clarified how it leads to "this paper uses the survey data...".

Thanks to the reviewer's suggestion, we have revised the text to express" the moderating effect of guardians' social participation" . The necessary changes have been implemented throughout the text. The modified parts are marked in red（lines 15）.

Specifying Heterogeneity Tests:

Expand a bit on the "Heterogeneity tests" to explain why these tests were conducted and what was specifically measured.

The core function of the "Heterogeneity tests" is to explore the differences between the variety of the groups. The results of inter group differences revealed that the effects of care types on orphan development are influenced by the gender and age of orphans. The modified parts are marked in red（lines 18-19）.

Introduction

1. In lines 30-32, that is According to China's relevant policies and regulations, the placement types for these orphans mainly include kinship care, institutional care, family foster care and legal adoption. Can you further explain what is meant by kinship care, institutional care, family care and legal adoption.

We sincerely appreciate the reviewer's thorough examination of our article. According to laws and cultural customs, the care types for these orphans mainly include kinship care, institutional care, family foster care and legal adoption. 

Kinship care occurs when close relatives of the orphan, such as grandparents, siblings, or other extended family members, voluntarily take on the responsibility of raising the orphan when the parents are deceased or unable to provide care. 

Institutional care refers to placing the orphan in a child welfare institution (such as an orphanage) established by the government or social organizations, where professionals provide daily care and education. This method is suitable when there are no relatives available to care for the orphan or when the relatives are unable to provide care.

Foster care involves placing the orphan in a foster family that has been carefully screened and trained, where foster parents provide care on behalf of the state. This method aims to provide the orphan with a family-like environment, promoting their psychological well-being and social adaptability.

Legal adoption refers to eligible families formally adopting the orphan through legal procedures, becoming their legal parents. After adoption, the adoptive family assumes the legal responsibility for the orphan's upbringing and education, and the orphan enjoys the same rights as biological children of the adoptive parents.

2. State the main objectives of the research. What does the researcher aim to achieve with this study?

We thank the reviewer for pointing out this. The question you raised is crucial to the article. 

In 2022, China had more than 190,000 orphans. These children were placed in four different types of care. Each care types have different effects on the development of orphans. The main objective of the study is to examine the relationship between care types and development of orphans in China. Furthermore, we want to explore what factors are influencing the effectiveness of orphan care.

3. Explain why this issue is important and identify the gaps or shortcomings in previous research that this study aims to address.

We appreciate your thoughtful inquiry. As we mentioned in the paper, in China, orphan care includes both relative and non-relative forms of parenting, as well as family-based and institutional-based care. Different care types have different effects on the development of orphans. However, there are few studies evaluating the effectiveness of different care types. Therefore, this paper tends to explore the impact of the care types on the development of orphans. Besides，the gaps or shortcomings in previous research have primarily centered on two key aspects. First, there is a lack of sufficient academic discussion on the factors contributing to the varying effects of different care types. Second, academic debate continues regarding whether different care types have distinct impacts on children across various groups.

We have supplemented and explained your suggestions regarding the introduction in this revised manuscript. We have modified the manuscript and also marked it in red in the manuscript (lines 76-77).

Discussion

1. In the discussion, avoid writing the table as in Table 4 Impact of placement type on the development of orphans of different genders and education levels

We thank the reviewer for this constructive suggestion, which helped us enhance the quality of the article. In response to your review comments, we have made structural adjustments to the content related to Table 4. We have incorporated most of the revised content into the conclusion section to present the results of the heterogeneity analysis by gender and age. Additionally, we have expanded the discussion section to further explore the relationship between our findings and previous academic research. The specific adjustments have been marked in red (lines 227-229，234-238，273-282) in the manuscript.

2. Discuss how your results relate to findings from previous relevant research. Identify any similarities, differences or contradictions between your results and previous research.

We sincerely appreciate the reviewer's thorough examination of our article. This paper supports the exploration of the relationship between care types and orphan development. We examined the impact of different care types on orphan development, as well as other factors influencing their growth across various care settings and the differences among them. The research findings indicate three parts. First, the results of this study show that the social participation of guardians has a moderating effect on the influence of care types on the development of orphans. Social participation plays an important role in the social isolation and vulnerable group of orphans and can promote the change of family resilience (Lin,2018). Therefore, when grandparents' social participation is increased, its positive impact on the development of orphans will also be strengthened. Second, compared with boys, care types has a greater impact on girls' development. The reason, combined with existing studies, may be explained as follows: on the one hand, girls care more about emotional quality than boys, their emotional needs are more sensitive than boys (Wang Su et al., 2020). Third, for orphans at different stages of education, the influence of care tpyes has obvious heterogeneity. The significant association between grandparents and child development occurs at a young age, and the influence of guardians on orphans diminishes with age (Li et al., 2022). Therefore, children in compulsory education are more vulnerable to grandparents and their social participation than children in non-compulsory education (Lin,2018). The specific adjustments have been marked in red (lines 227-229，234-238，273-282) in the manuscript.

Conclusions

The conclusion has clearly presented the main findings of the research. However, it needs to be revised to improve clarity and consistency to make it easier for readers to understand the main findings and implications of your research.

We thank this suggestion from the reviewer, which makes the conclusion more clearer. Following the principles of clarity, consistency, and other guidelines you suggested, we have rewritten the entire conclusion section. The new conclusion section is as follows:

By studying the impact of care types on the development of orphans in China, we found that grandparent care can promote the development of orphans. We also found that this impact was moderated by the social participation of guardians. The higher the social participation of guardians, the greater the positive impact of grandparent care on the development of orphans. Moreover, empirical research suggested that girls and younger orphans tend to exhibit better development in a grandparent care family.

Grandparent care is the predominant type of care for orphans’ care in China. Considering the economic situation and labor capacity of grandparents, more support, such as subsidies, should be provided to families where grandparents are the primary caregivers. The government should also ensure the implementation of relevant policy documents on orphan placement and protect the rights of grandparents as primary guardians.

The specific adjustments have been marked in red (lines 333-341) in the manuscript.

Once again, we appreciate the constructive guidance from the reviewer, and we are committed to implementing these valuable insights to enhance the clarity and focus of our paper.

---

## [Editor Report · Decision Letter 1]

16 Aug 2024

The influence of care types on the Development of Orphans——An empirical study from China

PONE-D-24-15042R1

Dear Dr. Feng Peng,

We’re pleased to inform you that your manuscript has been judged scientifically suitable for publication and will be formally accepted for publication once it meets all outstanding technical requirements.

Kind regards,

Satabdi Mitra, M.D(Community Medicine )

Academic Editor

PLOS ONE
---

## [Editor Report · Acceptance letter]

21 Aug 2024

PONE-D-24-15042R1 

PLOS ONE

Dear Dr. Feng, 

I'm pleased to inform you that your manuscript has been deemed suitable for publication in PLOS ONE. Congratulations! Your manuscript is now being handed over to our production team.

Kind regards, 

on behalf of

Dr Satabdi Mitra 

Academic Editor

PLOS ONE